# Morphologies of Wolf–Rayet Planetary Nebulae Based on IFU Observations

Ashkbiz Danehkar 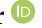

Department of Astronomy, University of Michigan, Ann Arbor, MI 48109, USA; danehkar@umich.edu

**Abstract:** Integral field unit (IFU) spectroscopy of planetary nebulae (PNe) provides a plethora of information about their morphologies and ionization structures. An IFU survey of a sample of PNe around hydrogen-deficient stars has been conducted with the Wide Field Spectrograph (WiFeS) on the ANU 2.3-m telescope. In this paper, we present the H$\alpha$ kinematic observations of the PN M 2-42 with a weak emission-line star (*wels*), and the compact PNe Hen 3-1333 and Hen 2-113 around Wolf–Rayet ([WR]) stars from this WiFeS survey. We see that the ring and point-symmetric knots previously identified in the velocity [N II] channels of M 2-42 are also surrounded by a thin exterior ionized H$\alpha$ halo, whose polar expansion is apparently faster than the low-ionization knots. The velocity-resolved H$\alpha$ channel maps of Hen 3-1333 and Hen 2-113 also suggest that the faint multipolar lobes may get to a projected outflow velocity of $\sim$100 $\pm$ 20 km s$^{-1}$ far from the central stars. Our recent kinematic studies of the WiFeS/IFU survey of other PNe around [WR] and *wels* mostly hint at elliptical morphologies, while collimated outflows are present in many of them. As the WiFeS does not have adequate resolution for compact ($\leq$6 arcsec) PNe, future high-resolution spatially-resolved observations are necessary to unveil full details of their morpho-kinematic structures.

**Keywords:** planetary nebulae; Wolf–Rayet stars; morphology; integral field spectroscopy



## 1. Introduction

Asymptotic giant branch (AGB) stars with low to intermediate initial masses (1–8 M$_\odot$) expel hydrogen-rich envelopes, which are subsequently photo-ionized by UV radiation from post-AGB degenerate cores, resulting in so-called *planetary nebulae* (PNe). The gaseous structures of these expanding ionized H-rich envelopes seen in the optical band, thanks to photoionization, are modified by stellar winds from their remnant central stars as they move on their evolutionary path toward the white dwarf stage. Their strong collisionally excited line emissions reveal their morphological features to us across the Galaxy, making them powerful kinematic probes. Aside from their morphologies, PNe also shed light on mass-loss processes occurring during the AGB phase and their transition to PN (see, e.g., [1–3]), as well as chemical elements produced by nucleosynthesis processes in the AGB stage (e.g., [4–6]).

A major challenge to theories in nebular astrophysics is the fact that many of them possess aspherical morphologies (see review by Balick and Frank [7]). Although single stars can generate aspherical shells, as predicted by the interacting stellar wind (ISW) theory [8] and its generalization [9], they cannot produce the highly complex elliptical and bipolar morphologies seen in recent high-resolution observations of many PNe (see e.g., [10]). It has also been proposed that binarity [11–13] and magnetic fields combined with stellar winds from a rotating single star [14,15] could also lead to aspherical morphologies. However, a single AGB star with realistic rotating speeds failed to generate a highly bipolar shell in magnetohydrodynamic simulations [16]. According to observations [17–20] and hydrodynamic simulations [21–23], the majority of aspherical morphologies appear to have formed via a binary channel.

A number of integral field unit (IFU) spectrographs on modest telescopes, such as the Wide Field Spectrograph (WiFeS [24,25]) on the Australian National University (ANU)

2.3-m telescope have made it possible to constrain the kinematic and ionization properties of several PNe [26–32]. Moreover, the Multi Unit Spectroscopic Explorer (MUSE) and the VIsible Multi-Object Spectrograph (VIMOS) on the European Southern Observatory's (ESO) Very Large Telescope (VLT) have recently enabled the constraint of the ionization structures of some PNe [33–36]. IFU spectroscopy allows us to spatially resolve the physical and kinematic properties of PNe, which is an important step toward understanding their morphological features in different ionization stratification layers.

An IFU survey of a sample of PNe around Wolf-Rayet ([WR]) central stars and weak emission-line stars (*wels*) were carried out with the ANU/WiFeS in April 2010 (Program No. 1100147; PI: Q.A. Parker). In particular, hydrogen-deficient [WR] stars constitute a substantial proportion (∼25%) of central stars of PNe, which demonstrate fast expanding atmospheres and high mass-loss rates [37], as well as helium-burning products (He, C, O, and Ne) on their stellar surfaces [4]. The WiFeS survey with a field-of-view of $25 \times 38$ arcsec$^2$ and a spatial resolution of $1''$ provides a wealth of information about their kinematic features [26–30], as well as their physical and chemical properties [38,39]. The WiFeS offers a seeing of $\sim 2''$ and a velocity resolution of $\sim$21 km s$^{-1}$ in the red channel at $R \sim$7000. This sample includes PNe around [WR] stars ranging from [WO 1] to [WC 6], and from [WC 9] to [WC 11] [37,40], as well as *wels* [41,42] (see Table 1 in Danehkar [29]), which could help us understand better the formation of aspherical morphologies and the mechanism scraping off hydrogen-rich layers in hydrogen-deficient degenerate cores. However, IFU spectroscopy and photoionization modeling of 4 PNe with supposed *wels* suggested that the *wels* classification could be spurious, since some assumed stellar emission lines could be of nebular origin, so three of them were then classified as hydrogen-rich O(H)-type [43]. In the present study, we provide the WiFeS H$\alpha$ kinematic results of the PN M 2-42 with a *wels* (Section 2), the PNe Hen 3-1333 and Hen 2-113 with [WC 10] stars (Section 3), followed by a discussion about our recent morpho-kinematic analyses of PNe around [WR] stars and a few *wels* conducted using the WiFeS survey [29] in Section 4, and a conclusion in Section 5.

## 2. M 2-42: H$\alpha$ Velocity Channels

The PN M 2-42 (=PNG008.2 − 04.8) is associated with a *wels* [41]. This object, which has a relatively high density ($3 \times 10^3$ cm$^{-3}$) and chemical abundances of a bit above the solar composition [39,44], could be in the Galactic bulge according to $D = 9.44$ kpc [45], as well as $7.4 \pm 0.6$ kpc derived from the surface brightness-radius correlation [46]. It contains two bipolar outflows extending from a dense central ring as revealed by long-slit [47] and IFU observations [28]. A three-dimensional (3D) morpho-kinematic model of this object was built based on the [N II] velocity-resolved channels [28], which apart from asymmetric outflows is roughly similar to the symmetric outflow model adopted by Akras and López [47]. The IFU observations disclosed highly asymmetric features of bipolar outflows, which could be created because of the nebular interaction with the interstellar medium (ISM).

Figure 1 shows a sequence of 10 velocity-resolved continuum-subtracted H$\alpha$ flux maps of M 2-42 on a logarithmic scale collected with channel intervals of around 21 km s$^{-1}$. The central velocity of each channel is given at the top of the panel with respect to the LSR (local standard of rest) systemic velocity ($v_{\mathrm{sys}} = 123$ km s$^{-1}$) listed at the right-bottom corner of the whole panels. The systemic velocity was transferred to the LSR frame using the IRAF function rvcorrect. The gray contours in each panel show the boundary at 10% of the mean H$\alpha$ surface brightness based on the H$\alpha$ plate retrieved from the SuperCOSMOS H-alpha Sky Survey (SHS) [48].

A dense central ring along with a pair of detached bipolar outflows were identified by Danehkar et al. [28] in the [N II] $\lambda$6584 channel maps. Additionally, the H$\alpha$ $\lambda$6563 velocity channels show that these structures are enveloped by a faint H$\alpha$ halo, which makes it difficult to observe the separation between the detached knots and central ring as seen in low-excitation [N II] maps. The H$\alpha$ flux maps are rather similar to what we see in the SHS H$\alpha$ image (see Figure 1 in Danehkar et al. [28]). We should note that the H$\alpha$ IFU maps

were saturated over two bright points on the main shell area (as previously pointed out by Danehkar et al. [28]). However, the Hα flux channels can help us identify the faint halo surrounding the bipolar outflows and the ring. Similarly, the PNe NGC 6567, NGC 6578, NGC 6629, and Sa 3-107 around *wels* were also found to contain large faint halos in the Hα emission, which are not visible in the [N II] maps [29].

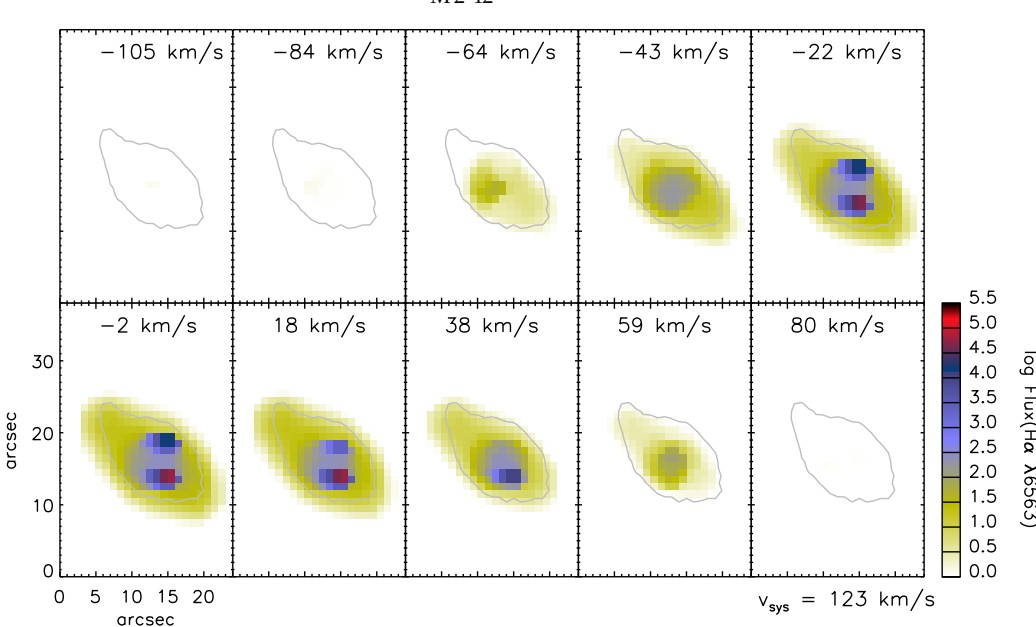

**Figure 1.** Velocity-resolved flux channels of M 2-42 along Hα $\lambda$6563 at $\sim$21 km s$^{-1}$ intervals with channel velocities specified at the top in km s$^{-1}$. The systemic velocity ($v_{sys}$) in the LSR frame is given in the right bottom corner in km s$^{-1}$ unit. The logarithmic color bar is in $10^{-15}$ erg s$^{-1}$ cm$^{-2}$ spaxel$^{-1}$ unit. The gray contour depicts the boundary of 10 % of the mean Hα surface brightness of this object in the SHS. The channels are oriented with north up and east toward the left side. Two bright points over the central shell in the flux maps are associated with Hα emission saturation.

The knots have a projected velocity of $\sim$15 km s$^{-1}$ with respect to the central star in the [N II] emission [47], which corresponds to an outflow velocity of $\sim$110 km s$^{-1}$ at the inclination of 82° adopted by Danehkar et al. [28]. However, the halos around the knots seem to expand with higher polar velocities in the Hα emission. The higher outflow velocities seen in the Hα channels could mostly be associated with the tenuous halos around the two detached N$^+$ low-ionization point-symmetric knots escaping from the core with slower outflow velocities. This could be a sign of the deceleration of the knots because of the interaction with the ambient medium. Similarly, the point-symmetric knots in Hb 4 were found to be decelerated by the PN-ISM interaction [29,49].

### 3. Hen 3-1333 and Hen 2-113: Hα Velocity Channels and PV Diagrams

The PNe Hen 3-1333 (=PNG 332.9 − 09.9) and Hen 2-113 (=PNG 321.0 + 03.9) are around late-type [WC 10] cool stars, according to the classification scheme of [40], with effective temperatures of 25 kK and 29 kK [50], respectively. These objects have extremely high densities of $10^5$ cm$^{-3}$ [39], which are typical of very young compact PNe. They also depict distinct oxygen- and carbon-rich dust characteristics of the early PN phase just after the AGB phase [51,52]. Their morphological features have been studied using the spatially-resolved Hα maps [27], which disclosed their main orientations on the sky plane, in agreement with the *Hubble Space Telescope* (*HST*) imaging analyses [53,54].

In Figure 2, we present the 15 flux channel maps of Hen 3-1333 and Hen 2-113 along the Hα emission on a logarithmic scale recorded at $\sim$21 km s$^{-1}$ velocity intervals. The channel velocity is with respect to the systemic velocity in the LSR frame provided at the right bottom corner. The continuum of each object was identified and removed from

each flux channel map. Similar to Figure 1, the contours again correspond to their SHS Hα distribution, which might illustrate the nebular borders. However, the SHS maps of these two objects are largely contaminated by the point spread function (PSF) of their central stars, resulting in the PSF spikes and artifacts around the PNe. It can be seen that the tenuous multipolar lobes extending from the central bodies in Hen 3-1333 and Hen 2-113 may reach a projected expansion velocity of about $\sim 100 \pm 20$ km s$^{-1}$ relative to their central stars. Previously, they found no evidence of Balmer line emission of stellar origin in these objects [50,55], so the Hα emission profiles could be attributed to nebular ionization structures. Nevertheless, the broad Hα wings in young, compact PNe may also be created by Rayleigh-Raman scattering [56]. We should note that the mass-weighted expansion velocities estimated from the half width at half maximum (HWHM) of the Hα emission in the integrated spectra were found to be 32 km s$^{-1}$ (Hen 3-1333) and 23 km s$^{-1}$ (Hen 2-113) [27], which are mostly associated with the bright central regions. However, the velocity channel maps indicate that the outflow velocities of the faint multipolar lobes may be much higher than the mass-weighted HWHM expansion velocities. Moreover, the spatially-resolved velocity maps previously built with a single Gaussian fitting only revealed the primary orientations of the bright nebulae [27], without any details about faint outflows.

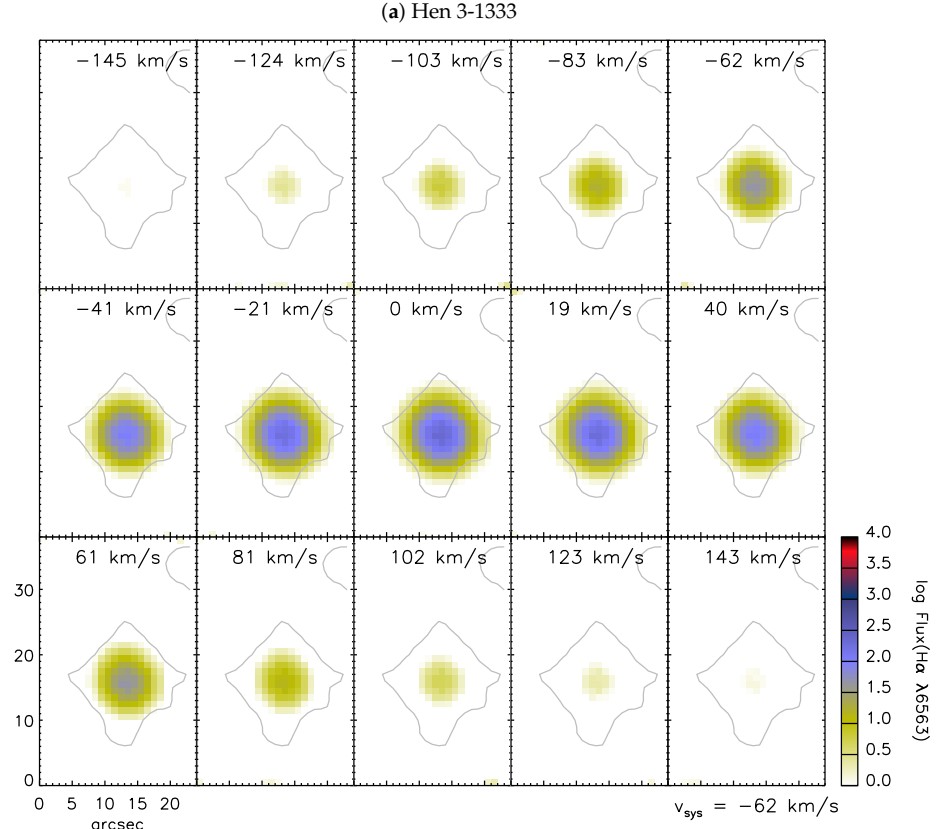

**(a)** Hen 3-1333

**Figure 2.** *Cont.*

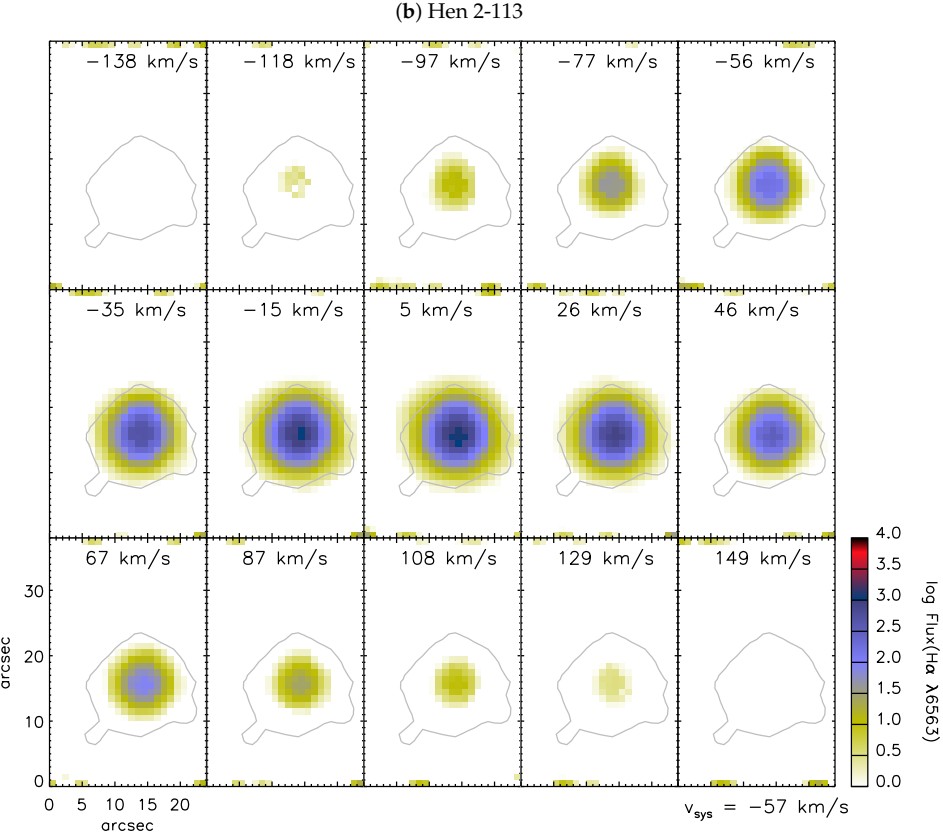

**Figure 2.** The same as Figure 1, but for (**a**) Hen 3-1333 and (**b**) Hen 2-113.

Figure 3 depicts the position-velocity (PV) arrays of Hen 3-1333 (top panels) and Hen 2-113 (bottom panels) observed in the H$\alpha$ $\lambda$6563 emission, which were generated with two slits with different orientations: one parallel with (left panel), and another one perpendicular to the symmetric axes of the primary orientations (right panel). The main on-sky orientations of the PN Hen 3-1333 and Hen 2-113 determined by Danehkar and Parker [27] have position angles (PA) of $-15°$ and $65°$, respectively, which are adopted for the slit directions. The slits of each object were positioned on the central star. The continua in the PV arrays were also detected and subtracted in a way similar to the continuum-subtraction performed in the velocity channel maps. The position and velocity axes are with respect to the central star's location and the object's systemic velocity, respectively. As can be seen, the faint outflows extending from the central nebulae may get as far as projected velocities of $\sim$100 km s$^{-1}$ similar to the channel maps shown in Figure 2, while they are also extended to a radial distance of $\sim$4″ from the central stars. Moreover, the tenuous lobes might be extended to $\sim$8″, where the nebula emission is extremely weak for detection. Asymmetric patterns seen in the horizontal slits in both of them, but rather symmetric patterns in the vertical slits, also indicate that the faint outflows have inclinations relative to the line of sight. The inclination angles of $i = -30°$ and $40°$ were also suggested for Hen 3-1333 and Hen 2-113 [27]. Detection of substructures of these compact objects require spatial and velocity resolution higher than the WiFeS.

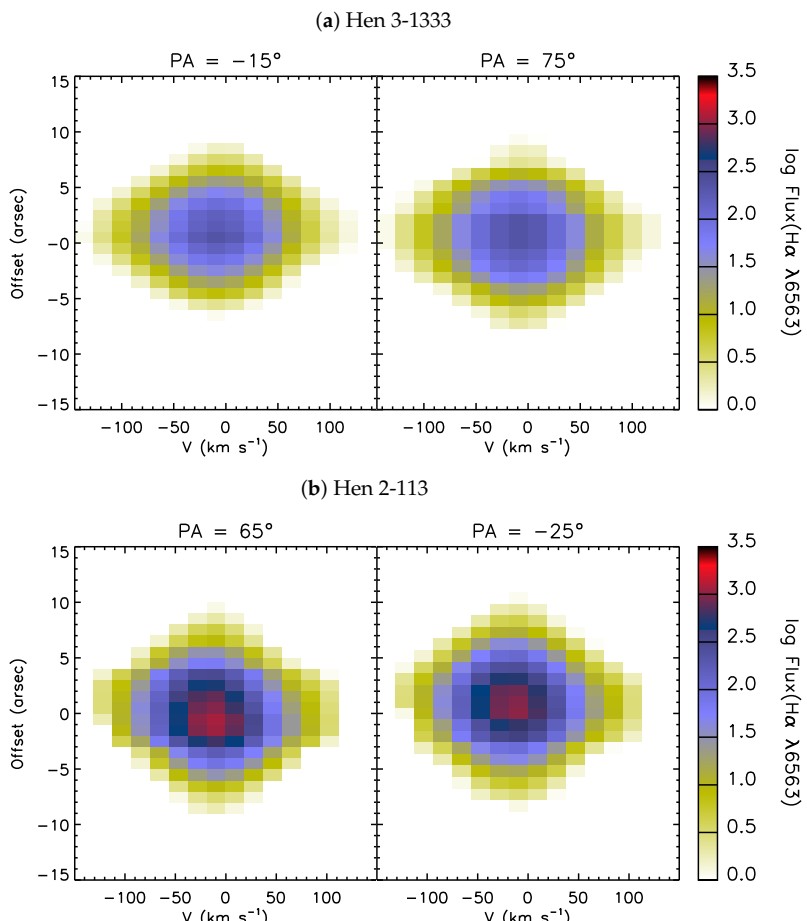

**Figure 3.** PV arrays of (**a**) Hen 3-1333 and (**b**) Hen 2-113 observed in the Hα λ6563 emission. Slits are aligned with the PA parallel to and vertical to the model's symmetric axis. The velocity in each P–V array is relative to the object's systemic velocity in km s$^{-1}$ units. The central star is positioned at the angular offset of $0''$. The logarithmic color bar is in $10^{-15}$ erg s$^{-1}$ cm$^{-2}$ spaxel$^{-1}$ unit.

## 4. Discussion: Morpho-Kinematics of [WR] PNe

A sample of PNe surrounding hydrogen-deficient [WR] stars, including the early-type group ranging from [WO 1] to [WC 6], and the late-type group from [WC 9] to [WC 11] [37,40], as well as a few *wels* [41,42], have been studied by Danehkar [29], which might provide some clues about aspherical morphologies and hydrogen-deficient central stars. The projected velocity maps, PV diagrams, velocity channel maps, and archival images have been used to constrain their 3D geometrical models in the SHAPE program [57,58].

Figure 4 presents their wireframe SHAPE models (top panel) together with their corresponding Doppler shift maps generated by SHAPE (bottom panel). Redshift and blueshift effects with respect to the nebula center are rendered by red and blue colors, which can be compared to the two-dimensional (2D) projected radial velocity IFU maps (see Figure 1 in Danehkar [29]). Particularly, PV arrays and velocity channels created by SHAPE put constraints on the geometry with specified density and velocity laws, whereas Doppler shift rendered images (Figure 4 bottom) help us have a first-order approximation of the morphology. The final 3D models, which were built with PV diagrams and velocity-resolved channels, are presented in an interactive figure in Danehkar [29], together with the 3D object files on the Sketchfab platform (https://skfb.ly/opFZv, 29 January 2022).[1]

Wireframe SHAPE Models

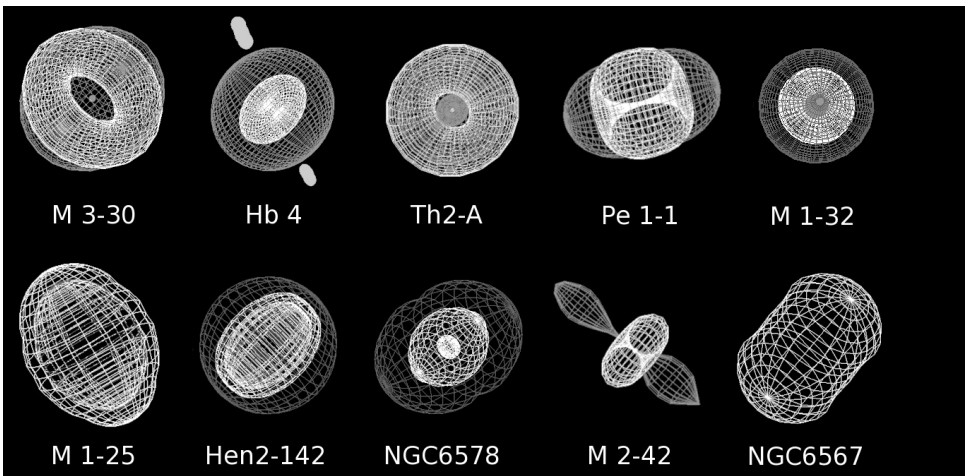

Doppler Shift SHAPE Outputs

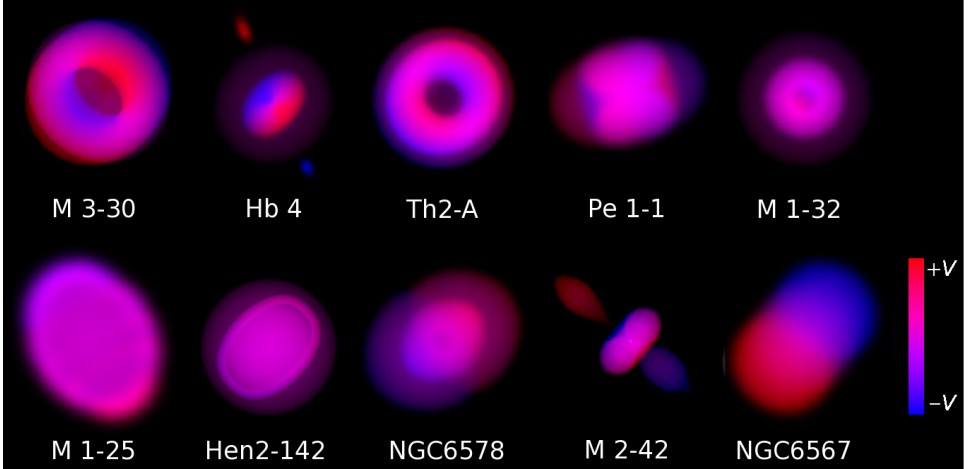

**Figure 4.** Wireframe SHAPE models (**top panel**) and their Doppler shift outputs (**bottom panel**) of M 3-30, Hb 4, Pe 1-1, M 1-32, M 1-25, Hen 2-142, MGC 6578, and NGC 6567 from Danehkar [29], along with Th 2-A [26], and M 2-42 [28]. Red and blue colors in the Doppler shift outputs are associated with redshift ($+V$) and blueshift ($-V$) effects relative to the centers, respectively. 3D interactive SHAPE models are provided by Danehkar [29], hosted on Sketchfab (https://skfb.ly/opFZv, 29 January 2022), and stored in a Zenodo data repository (doi:10.5281/zenodo.5393974).

As can be seen, these PNe mostly have aspherical, elliptical morphologies. Moreover, collimated bipolar outflows can be easily distinguished in the position-position-velocity (PPV) cubes of Hb 4 and M 2-42 as shown in Figure 5 (see [29] for velocity slices of Hb 4, and Figure 1 for M 2-42). Collimated outflows expanding from the ring-shaped main structures were also detected in the observed PV arrays of M 1-32, M 3-15, M 3-30 [29], and Th 2-A [26]. We should also mention prolate spheroid shells in NGC 6567, NGC 6578, NGC 6629, Hen 2-142, and M 1-25. Additionally, as pointed out by Danehkar [29], the IFU projected dispersion maps of Pe 1-1 hint at possible collisions between the bipolar outflow and the previous expelled materials. The kinematic ionization structure of M 2-42 was previously determined from the [N II] spatially-resolved IFU maps [28], which are now complemented by the velocity-resolved H$\alpha$ flux maps in Section 2. Hen 3-1333 and Hen 2-113 around [WC 10], whose velocity channels and PV arrays are presented in Section 3, were also studied in detail by Danehkar and Parker [27] using the 2D projected velocity IFU maps and high-resolution *HST* images that pointed to multipolar outflows and tenuous lobes extending from their central, dense ($\sim 10^5\,\mathrm{cm}^{-3}$ [39]) bodies.



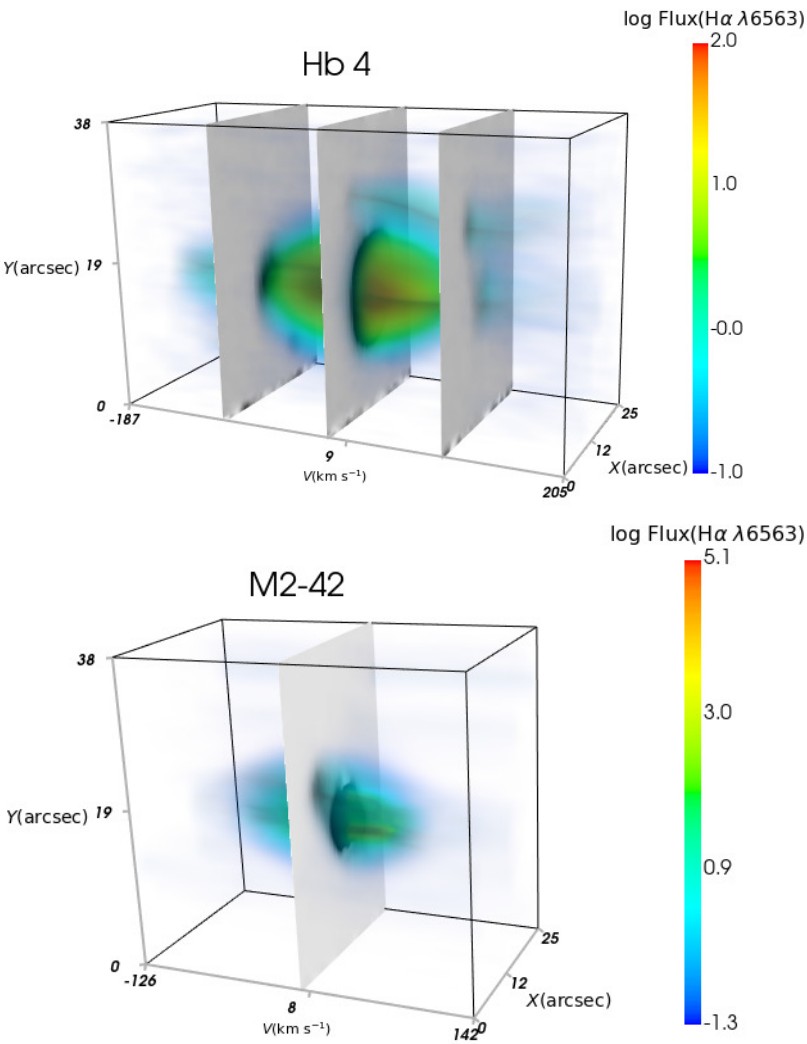

**Figure 5.** PPV cubes of Hb 4 (**top panel**) and M 2-42 (**bottom panel**) with 3 slices at $-104$, 0, and $+102$ km s$^{-1}$ for Hb 4, and a slice at $-2$ km s$^{-1}$ for M 2-42. Velocity slices of Hb 4 and M 2-42 are presented in Danehkar [29] and Figure 1, respectively.

## 5. Conclusions

IFU observations with cost-effective spectrographs on modest telescopes such as the ANU/WiFeS can disclose detailed morpho-kinematic properties of extended Galactic PNe (e.g., M 2-42 in Section 2), while they are also promising for compact ($\leq 6$ arcsec) PNe (e.g., Hen 3-1333 and Hen 2-113 in Section 3). There are various kinematic information that can be extracted from an IFU datacube (see, e.g., Figure 5), namely spatially-resolved projected velocity maps, velocity-resolved flux channels, and PV diagrams.

The H$\alpha$ channel maps of M 2-42 in Section 2 indicate that the previously identified ring and point-symmetric knots [28,47] are also covered by a low-density exterior H$^+$ halo, which cannot easily be identified with low-excitation N$^+$ mapping. The polar regions in the exterior H$^+$ halo expand (see Figure 1) apparently higher than those in the point-symmetric knots ($\sim 110$ km s$^{-1}$). As point-symmetric knots are typically expected to move faster than their surrounding halos, their slowness could be an indication of the deceleration due to a collision with the surrounding medium (see also Hb 4 in [29,49]).

The velocity channels and PV diagrams of Hen 3-1333 and Hen 2-113 in Section 3 also revealed that the faint outflows may approach a projected outflow velocity of $\sim 100 \pm 20$ km s$^{-1}$ at $4''$ far from the central stars. However, we caution that the broad H$\alpha$ wings of these compact PNe may be associated with Rayleigh-Raman scattering [56]. Asymmetric PV patterns seen in the horizontal slits could also be related to moderate line-of-sight inclinations

($i = -30°$ and $40°$) in Hen 3-1333 and Hen 2-113. Detailed morpho-kinematic modeling of these compact PNe can be performed when kinematic observations with higher spatial and wavelength resolution are available.

The on-sky orientation of an extended symmetric nebula can easily be determined from a spatially-resolved 2D radial velocity IFU map (see Figure 1 in [29]) by comparison with a Doppler shift synthetic image created by SHAPE (Figure 4 bottom), but without full details of substructures. 3D morpho-kinematic modeling implemented using velocity-resolved channel maps and position-velocity diagrams may disentangle different structural components of an ionized nebula, which cannot be unveiled by 2D radial velocity IFU mapping.

**Funding:** This research was partially supported by a Macquarie Research Excellence Scholarship (MQRES: 2010–2013) and a Sigma Xi Grants-in-Aid of Research (GIAR; 2013).

**Data Availability Statement:** The supporting data for Section 4 are provided as supplementary materials in Danehkar [29]. The 3D models are available on Sketchfab (https://skfb.ly/opFZv, 29 January 2022) and stored in a Zenodo data repository (doi:10.5281/zenodo.5393974). The data underlying Sections 2 and 3 will be shared on reasonable request to the author.

**Acknowledgments:** The author thanks the APN8 2021e organizers for providing an opportunity to present this work, Quentin Parker and David Frew for supporting the WiFeS observing run in 2010, Milorad Stupar for instructions on the WiFeS data reduction, and Kyle DePew for undertaking the 2010 WiFeS observation.

**Conflicts of Interest:** The author declares no conflict of interest.

## Abbreviations

The following abbreviations are used in this manuscript:

| | |
|---|---|
| ANU | Australian National University |
| HST | Hubble Space Telescope |
| HWHM | Half Width at Half Maximum |
| IFU | Integral Field Unit |
| ISM | Interstellar Medium |
| LSR | Local Standard of Rest |
| PA | Position Angle |
| PN | Planetary Nebula |
| PSF | Point Spread Function |
| PPV | Position-Position-Velocity |
| PV | Position-Velocity |
| SHS | SuperCOSMOS H-alpha Sky Survey |
| *wels* | weak emission-line star |
| WiFeS | Wide Field Spectrograph |
| WR | Wolf–Rayet |

## Note

1. Archived in a Zenodo repository: https://doi.org/10.5281/zenodo.5393974.

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
