# Peer review of "Morphologies of Wolf–Rayet Planetary Nebulae Based on IFU Observations"

_galaxies, doi:10.3390/galaxies10020045_

Round 1
Reviewer 1 Report
This paper presents a brief morphologic and kinematic study of three compact planetary nebulae (PNs) based on the spectroscopic results for WiFeS/IFU data. The
application of WiFeS/IFU spectroscopic data is important for understanding the kinematic information and morphologic distribution of studied PNs. However, some
key issues are not clear and similar spectroscopic results had been published in previous studies (ex: Danehkar et al., 2016, AJ, 151, 38). This draft with
current version is needed to be clarified and is far to publish. Some main comments and suggestions are listed below:
Main comments:
The author has presented the kinematic studies of these PNs based on IFU spectroscopic data with [NII] line in previous papers already, why the author report similar
results in H-alpha emission for the same objects repeatedly? What is the purpose of the author? These structures observed in H-alpha line are always mixed with outer
materials ejected from early stages of PNs.
Section 1. Introduction:
1. Paragraph 4:
"The WiFeS results of a sample of PNe around Wolf-Rayet ([WR]) central stars have .....in Danehkar".
It is unclear for readers why the WR PNs are important for us? Why we want to study this type of PNs? Are they unique among PNs? Is there any difference between WR, wel,
and normal PNs?
Section 2. Kinematic structures of PNe with [WR] stars:
1. Paragraph 1:
"IFU data of a sample of PNe around [WR]-type central stars and a weak emission line star (wels) were collected with the ANU/WiFeS in April 2010".
Why the author select these PNs as a sample? Is there any selection effect for this sample of WR PNs?
2. Paragraph 3:
"Detached point-symmetric knots can be easily distinguished in the position-position-velocity (PPV) cubes of Hb 4 and M 2-42 as shown in Fig. 2."
The structures named as "point-symmetric knots" of M 2-42 do not look like "knotty" in the figures 1 and 2. They show a lobe-like appearance.
For avoiding further misunderstanding, it will be better to modify the names of these structures.
Section 3. M 2-42: H-alpha velocity channels:
1. Paragraph 4:
"The fast outflow velocities seen in the H-alpha channels could mostly be associated with the tenuous halo around the two detached N+ low-ionization
point-symmetric knots escaping from the core with slower outflow velocities".
As usual, the velocities of these knots with [NII] line are faster than the ones of their surrounding structures such as halo. What mechanism makes
these knots slower than the tenuous halo?
Section 4. Hen 3-1333 and Hen 2-113: H-alpha velocity channels and PV diagrams
1. Paragraph 2:
"It can be seen that the tenuous multipolar lobes extending from the central bodies in Hen 3-1333 and Hen 2-113 reach maximum expansion velocities of about
~ 120 km s^−1 relative to their central stars".
From figure 4, it seems that the maximum expansion velocity of about ~ 140 km s^−1 relative to its central star for Hen 3-1333. In addition, how the author
is convinced of the maximum expansion velocities of about 120 km s^−1 relative to their central stars for "multipolar lobes"?
"We should note that the HWHM expansion velocities .... central regions".
What is the meaning of "HWHM"?
2. Paragraph 3:
"Asymmetric patterns seen in the horizontal slits ... the faint outflows have inclinations relative to the line of sight".
Do previous PV arrays with [NII] line show similar results? It is interesting that the author do a comparison between the results
with H-alpha and [NII] lines?
Section 5. Conclusions:
Paragraph 2:
"The polar regions in the exterior H+ halo are expanding with deprojected velocities .... higher than those in the point-symmetric knots".
Same as previous comments listed in section 3, these results make the readers confusing and the author should clarify them.
Figures:
1. Figure 1:
The displays with blue and red colors for Doppler shift outputs in this figure are unclear for readers without any gradient color.
2. Figure 4:
The obvious boundaries of H-alpha contour curves plotted from SHS data are found to be contaminated from some field stars after comparing SHS H-alpha and
HST with well resolved images, thus I confused whether the contour plots presented in the figure indeed reflect the true shapes of these two compact PNs?
In addition, the contaminations from asterisms of these bright PNs can also be seen in the SHS images. What is the reason why the author want to present the
contours here?
3. Figure 5:
The contours plotted overlaid on the PV arrays or make false-color plots are needed to better show spatial velocity distributions.
Author Response
RESPONSES TO THE REVIEWER'S COMMENTS:
COMMENT: This paper presents a brief morphologic and kinematic study of three compact planetary nebulae (PNs) based on the spectroscopic results for WiFeS/IFU data. The application of WiFeS/IFU spectroscopic data is important for understanding the kinematic information and morphologic distribution of studied PNs. However, some key issues are not clear and similar spectroscopic results had been published in previous studies (ex: Danehkar et al., 2016, AJ, 151, 38). This draft with current version is needed to be clarified and is far to publish. Some main comments and suggestions are listed below:
RESPONSE: I have carefully read the reviewer’s comments, and I am thankful to the referee for the helpful suggestions. I have revised the manuscript and addressed the referee's main issue (# 1 below). I also re-plotted PV arrays by a false-color scale as asked by the referee (#12 below), and revised our text (other points) based on the referee's comments. My responses to the referee's points are as follows:
1. COMMENT: Main comments:
The author has presented the kinematic studies of these PNs based on IFU spectroscopic data with [NII] line in previous papers already, why the author report similar results in H-alpha emission for the same objects repeatedly? What is the purpose of the author? These structures observed in H-alpha line are always mixed with outer materials ejected from early stages of PNs.
1. RESPONSE: This was now clarified in the abstract: “We see that the ring and point-symmetric knots previously identified in the [N II ] IFU velocity channels of the PN M 2-42 are also surrounded by a thin exterior ionized Hα halo, whose polar expansion is apparently faster than the low-ionization knots.” The [N II] observations correspond to low-ionization features. As shown in the recent works (https://arxiv.org/abs/2112.12043 and https://arxiv.org/abs/2107.03994 2nd version posted this week), the [N II] and H-alpha morphological features do not always have the simialr ionization layers due to shock-ionization or the presence of a faint H-alpha halo. This added in the revised version in Sec 2: “Similarly, the PNe NGC 6567, NGC 6578, NGC 6629, and Sa 3-107 around wels are also found to contain large faint halos in the Hα emission, which are not visible in [N II ] [29].”
2. COMMENT: Section 1. Introduction:
1. Paragraph 4:
"The WiFeS results of a sample of PNe around Wolf-Rayet ([WR]) central stars have .....in Danehkar".
It is unclear for readers why the WR PNs are important for us? Why we want to study this type of PNs? Are they unique among PNs? Is there any difference between WR, wel, and normal PNs?
2. RESPONSE: Added: “In particular, hydrogen-deficient [WR] stars constitute a substantial proportion ( ∼ 25%) of central stars of PNe, which demonstrate fast expanding atmospheres and high mass-loss rates [34], as well as helium-burning products (He, C, O and Ne) on their stellar surfaces [35]. … This sample includes PNe around [WR] stars ranging from [WO1] to [WC6], and from [WC9] to [WC11] [34,37], as well as wels [38,39] (see Table 1 in Danehkar [29]), which could help us understand better the formation of aspherical morphologies and the mechanism scraping off hydrogen-rich layers in hydrogen-deficient degenerate cores.”
3. COMMENT: Section 2. Kinematic structures of PNe with [WR] stars:
1. Paragraph 1:
"IFU data of a sample of PNe around [WR]-type central stars and a weak emission line star (wels) were collected with the ANU/WiFeS in April 2010".
Why the author select these PNs as a sample? Is there any selection effect for this sample of WR PNs?
3. RESPONSE: Text revised: “A sample of PNe surrounding hydrogen-deficient [WR] stars, including the early-type group ranging from [WO1] to [WC6], and the late-type group from [WC9] to [WC11 [34,37], as well as a few wels [38,39], have been studied by Danehkar [29], which might provide some clues about aspherical morphology and hydrogen-deficient central stars.”
4. COMMENT: 2. Paragraph 3:
"Detached point-symmetric knots can be easily distinguished in the position-position-velocity (PPV) cubes of Hb 4 and M 2-42 as shown in Fig. 2."
The structures named as "point-symmetric knots" of M 2-42 do not look like "knotty" in the figures 1 and 2. They show a lobe-like appearance. For avoiding further misunderstanding, it will be better to modify the names of these structures.
4. RESPONSE: Revised: “collimated bipolar outflows can be easily distinguished in the position-position-velocity (PPV) cubes of Hb 4 and M 2-42”
5. COMMENT: Section 3. M 2-42: H-alpha velocity channels:
1. Paragraph 4: "The fast outflow velocities seen in the H-alpha channels could mostly be associated with the tenuous halo around the two detached N+ low-ionization point-symmetric knots escaping from the core with slower outflow velocities".
As usual, the velocities of these knots with [NII] line are faster than the ones of their surrounding structures such as halo. What mechanism makes these knots slower than the tenuous halo?
5. RESPONSE: Added: “This could be a sign of the deceleration of the knots because of the interaction with the ambient medium. Similarly, the point-symmetric knots in Hb 4 were found to be decelerated by the PN-ISM interaction [29,48].”
6. COMMENT: Section 4. Hen 3-1333 and Hen 2-113: H-alpha velocity channels and PV diagrams
1. Paragraph 2:
"It can be seen that the tenuous multipolar lobes extending from the central bodies in Hen 3-1333 and Hen 2-113 reach maximum expansion velocities of about ~ 120 km s^−1 relative to their central stars".
From figure 4, it seems that the maximum expansion velocity of about ~ 140 km s^−1 relative to its central star for Hen 3-1333. In addition, how the author is convinced of the maximum expansion velocities of about 120 km s^−1 relative to their central stars for "multipolar lobes"?
6. RESPONSE: Velocity channel maps in Figure 2 suggest a maximum *projected* velocity of ~120 km relative to their central stars. In the revised paper, “projected” was added to text. A deprojected outflow velocity can be derived if we accurately know the inclinations of multipolar lobes.
7. COMMENT: "We should note that the HWHM expansion velocities .... central regions".
What is the meaning of "HWHM"?
7. RESPONSE: Added: “the half width at half maximum (HWHM)”
8. COMMENT: 2. Paragraph 3:
"Asymmetric patterns seen in the horizontal slits ... the faint outflows have inclinations relative to the line of sight".
Do previous PV arrays with [NII] line show similar results? It is interesting that the author do a comparison between the results with H-alpha and [NII] lines?
8. RESPONSE: While [N II] PV arrays of M 2-42 around a hot CS (Teff = 75 Kk) were used for a kinematic study (https://arxiv.org/abs/1601.01702), the [N II] data of the compact PNe Hen 2-142 and Hen 3-1333 with cool CSs (Teff = 30 Kk) may not be suitable for a detailed kinematic analysis and need further investigation, so we present only the H-alpha observations of these 3 PNe in this paper.
9. COMMENT: Section 5. Conclusions:
Paragraph 2:
"The polar regions in the exterior H+ halo are expanding with deprojected velocities .... higher than those in the point-symmetric knots".
Same as previous comments listed in section 3, these results make the readers confusing and the author should clarify them.
9. RESPONSE: Revised (see also Point #5 above). Text added to Sec 5: “As point-symmetric knots are typically expected to move faster than their surrounding halo, their slowness could be an indication of the deceleration due to a collision with the surrounding medium (see e.g. [29,48]).”
10. COMMENT: Figures:
1. Figure 1:
The displays with blue and red colors for Doppler shift outputs in this figure are unclear for readers without any gradient color.
10. RESPONSE: A gradient color bar showing +V and -V respect to the central star was added.
Noting that SHAPE program generates Doppler shift synthetic images in arbitrary scales, which can be used for quick comparison with projected radial velocity IFU maps.
11. COMMENT: 2. Figure 4:
The obvious boundaries of H-alpha contour curves plotted from SHS data are found to be contaminated from some field stars after comparing SHS H-alpha and HST with well resolved images, thus I confused whether the contour plots presented in the figure indeed reflect the true shapes of these two compact PNs?
In addition, the contaminations from asterisms of these bright PNs can also be seen in the SHS images. What is the reason why the author want to present the contours here?
11. RESPONSE: The counters for Hen 3-1333 and Hen 2-113 were produced in the same way made for M 2-42 showing the H-alpha emission distribution, which may show the nebular border. We now clarified the PSF problem with SHS for Hen 3-1333 and Hen 2-113 in text: “Similar to Figure 1, the contours again correspond to their SHS Hα distribution, which might illustrate the nebular borders. However, the SHS maps of these two objects are largely contaminated by the point spread function (PSF) of their central stars, resulting in the PSF spikes and artifacts around the PNe.””
12. COMMENT:3. Figure 5:
The contours plotted overlaid on the PV arrays or make false-color plots are needed to better show spatial velocity distributions.
12. RESPONSE: PV arrays are now presented by a false-color scale as requested.
Reviewer 2 Report
The paper presents basic results from a IFU studies of a number of planetary nebulae.
It is heavily dependent on papers [26] and [27]. [27] is I think published, but the status of [26] is unclear. It is available on arxiv but without any publication information, and may just be submitted but still under review.
Line 40-42. Muse and VIMOS lack the spectral resolution needed for kinematic studies. For that, see Gesicki et al. 2016, A&A 585, and the pioneering work by Reay et al. 1983, MNRAS 203 1087.
Line 63. One table is referred to with two different publications. Which one is correct?
Section 2
This section appears to be mainly an extract from reference [26]. It cannot be read on its own: for essential information and figures it refers to that paper. I counted 8 referrals to that paper, and I am still unsure what that paper actually concludes as there is only a rough description of structures without interpretation or conclusion. The description of the observations that is given in Section 2 is relevant, but the remainder seems to have no role in this paper.
Section 3
Fig. 3 The caption states '~ 10 average Ha surface brightness' . I don't have a clue what that means.
Lines 120-125 compare the Halpha structure to the [NII] reported in [25]. The statement is made here that the Halpha expands twice as fast. That is unclear. The image slices are taken at different velocities for Ha and [NII], which makes comparing difficult. The NII lobes have a half velocity width of at least 20 km/s, not 15 as stated here but they are also 10 times fainter than in Halpha, meaning that any difference is likely dominated by sensitivity. The thermal width of the Halpha line is also 3 times larger than that of NII. I cannot readily confirm the conclusion of the author.
Section 4. The main new result in this paper is the fast outflow in Hen 2-113. I have not come across this before. It is not visible in Fig. 4 and only shows up in Fig. 5. However, Fig. 5 is a poor reproduction which shows very little detail. My impression from this figure, especially the bar structure in the left panel, is that this is associated with the star rather than the nebula. The author argues for a 4-arcsec extension but this seems consistent with the 2" seeing (mentioned in section 2).
I note that the velocity width agrees with the known velocities of the stellar winds in these stars (de Marco et al). The paper does not give enough information to separate the star from the nebula, and therefore the reader can conclude that this feature traces the stellar wind rather than the nebula. [WC] stars can have Halpha lines, and can also have a HeII line coincident with Halpha.
In the conclusions, the paper does not return to the topic of [WC] stars. What does it tell us about such stars? How do these morphologies differ (or not) from those of other stars?
I recommend to remove most of section 2, to make the paper self-contained, and to better present the evidence for a fast nebular flow in Hen 2-113. Some minor improvements to the English are also needed.
Author Response
RESPONSES TO THE REVIEWER'S COMMENTS:
COMMENT: The paper presents basic results from a IFU studies of a number of planetary nebulae.
It is heavily dependent on papers [26] and [27]. [27] is I think published, but the status of [26] is unclear. It is available on arxiv but without any publication information, and may just be submitted but still under review.
RESPONSE: For your information, the paper [29] was submitted to ApJS in July 2021, and it was accepted last week. The final version was just posted to arXiv, which will appear on the arXiv next week. I am thankful to the referee for the helpful comments, and I have carefully addressed the reviewer’s comments below. Accordingly, I have revised the manuscript based on the referee's main points (#3, #7, and #8 below), and amended text to address the referee's comments as follows:
1. COMMENT: Line 40-42. Muse and VIMOS lack the spectral resolution needed for kinematic studies. For that, see Gesicki et al. 2016, A&A 585, and the pioneering work by Reay et al. 1983, MNRAS 203 1087.
1. RESPONSE: Revised: “enabled the constraint of the ionization structures”.
2. COMMENT: Line 63. One table is referred to with two different publications. Which one is correct?
2. RESPONSE: Corrected: “(see Table 1 in Danehkar [29])”
3. COMMENT: Section 2
This section appears to be mainly an extract from reference [29]. It cannot be read on its own: for essential information and figures it refers to that paper. I counted 8 referrals to that paper, and I am still unsure what that paper actually concludes as there is only a rough description of structures without interpretation or conclusion. The description of the observations that is given in Section 2 is relevant, but the remainder seems to have no role in this paper.
3. RESPONSE: This section was revised and moved to a Discussion Section Before the Conclusion in order to address Point #7 (below).
4. COMMENT: Section 3
Fig. 3 The caption states '~ 10 average Ha surface brightness' . I don't have a clue what that means.
4. RESPONSE: Revised: “depicts the boundary of 10 percent of the mean Hα surface brightness of this object in the SHS”.
5. COMMENT: Lines 120-125 compare the Halpha structure to the [NII] reported in [25]. The statement is made here that the Halpha expands twice as fast. That is unclear. The image slices are taken at different velocities for Ha and [NII], which makes comparing difficult. The NII lobes have a half velocity width of at least 20 km/s, not 15 as stated here but they are also 10 times fainter than in Halpha, meaning that any difference is likely dominated by sensitivity. The thermal width of the Halpha line is also 3 times larger than that of NII. I cannot readily confirm the conclusion of the author.
5. RESPONSE: Reference corrected: 15 km/s was measured by Akras & Lopez, MNRAS 2012, 425, Page 2200, they mentioned “The knots are observed to be expanding with a velocity of ∼± 15 km s−1 with respect to the core.” Our flux maps are shown on a logarithmic scale and with different maximum scales for Ha and [NII].
Text and conclusion revised: “The knots have a projected velocity of ∼ 15 km s − 1 respect to the central star in the [N II] emission [41], which corresponds to an outflow velocity of ∼ 90 km s − 1 at the inclination of 80 ◦. However, the halos around the knots seem to expand with a higher velocity in the Hα emission. The higher outflow velocities seen in the Hα channels could mostly be associated with the tenuous halo around the two detached N+ low-ionization point-symmetric knots escaping from the core with slower outflow velocities.”
6. COMMENT: Section 4. The main new result in this paper is the fast outflow in Hen 2-113. I have not come across this before. It is not visible in Fig. 4 and only shows up in Fig. 5. However, Fig. 5 is a poor reproduction which shows very little detail. My impression from this figure, especially the bar structure in the left panel, is that this is associated with the star rather than the nebula. The author argues for a 4-arcsec extension but this seems consistent with the 2" seeing (mentioned in section 2).
I note that the velocity width agrees with the known velocities of the stellar winds in these stars (de Marco et al). The paper does not give enough information to separate the star from the nebula, and therefore the reader can conclude that this feature traces the stellar wind rather than the nebula. [WC] stars can have Halpha lines, and can also have a HeII line coincident with Halpha.
6. RESPONSE: PV arrays of Hen 3-1333 and Hen 2-113 are now presented by a false-color scale to better show PV distributions as well as 4-arcsec extension. We also note that “The continua in the PV arrays were also detected and subtracted...”
These two PNe are around hydrogen-deficient [WC 10] stars with Teff=30 kK which are not capable of producing strong stellar and nebular He II emission. There is no report of He II 6560 stellar emission in De Marco et al (1997, 1998). Moreover, they mentioned that they “find no detectable amounts of hydrogen in either stellar wind” and and “The lack of evidence for Balmer line emission from the stellar winds” in De Marco et al (1997, MNRAS 292, 86), so there is no H-alpha stellar emission. Stellar winds of Hen 3-1333 and Hen 2-113 were also estimated using the He I PCyg profiles in De Marco et al (1998, MNRAS 296, 419). Moreover, the stellar emission lines listed in Table 7 and 8 in De Marco et al (1997, MNRAS 297, 999) do not show any stellar emission lines around the nebular H-alpha emission.
The above point was clarified in the revised version: “Previously, they found no evidence of Balmer line emission of stellar origin in these objects [46,51], so the Hα emission profiles could be attributed to nebular ionization properties. Nevertheless, the broad H-alpha wings in young, compact PNe may also be created by Rayleigh-Raman scattering [53].”
7. COMMENT: In the conclusions, the paper does not return to the topic of [WC] stars. What does it tell us about such stars? How do these morphologies differ (or not) from those of other stars?
7. RESPONSE: Please see our answer to Point #3 (above). This was specified in Sec 4, which a summary of the paper [29].
In the revised version, the following sentences were also added to the last paragraph in Introduction: “In particular, hydrogen-deficient [WR] stars constitute a substantial proportion ( ∼ 25%) of central stars of PNe, which demonstrate fast expanding atmospheres and high mass-loss rates [34], as well as helium-burning products (He, C, O and Ne) on their stellar surfaces [35]. … This sample includes PNe around [WR] stars ranging from [WO1] to [WC6], and from [WC9] to [WC11] [34,37], as well as wels [38,39] (see Table 1 in Danehkar [26]), which could help us understand better the formation of aspherical morphologies and the mechanism scraping off hydrogen-rich layers in hydrogen-deficient degenerate cores.”
8. COMMENT: I recommend to remove most of section 2, to make the paper self-contained, and to better present the evidence for a fast nebular flow in Hen 2-113. Some minor improvements to the English are also needed.
8. RESPONSE: Revised. Please see our answer to Points #3 and #7 (above).
Reviewer 3 Report
Referee's Report on:
A. Danehkar
Morphologies of Wolf-Rayet Planetary Nebulae based on IFU
Observations
This paper presents a summary of kinematic results with WiFeS already
published by the author and new data and analysis of H-alpha kinematics of
several of these [WR] PNe. The paper needs some minor improvements before
being acceptable for publication.
Major comments
--------------
l.62
Basurah et al. (2016, 2016MNRAS.458.2694B) have pointed out the possible
problems and ill definition of the wels category of PNe and this should be
mentioned.
l. 145 Sentence beginning "However, the velocity ..." and referring to
Figure 5.
Are the authors sure that these are high velocity line wings, indicative of
nebular expansion velocities > HWHM. From the figure, the referee was struck
that both PNe have similar width wings, so could these wings be the intrinsic
wings of the spectrograph line spread function. Alternatively are these
scattering wings of the intrinsic H-alpha emission of the PNe, such as
electron scattering, and not necessarily indicative of high velocity outflow.
It is not possible to judge how much fainter are the wings than the peaks
from Fig. 5, but examination of Fig. 4 suggests a factor > 1000. This point
should at least be mentioned.
Minor comments
--------------
Abstract
l.12 to read ... lobes which reach ...
The last sentence is not clear - what further light will higher spectral
resolution observations reveal given that the WiFeS observations already
have modest spectral resolution.
1. Introduction
l.19 Surely the Universe is H-rich. The interest of PNe is that they
can be enriched in heavier elements indicative of phases of nuclear
reprocessing on the AGB and PN tracks.
l.33 suggest "rotating" rather than "spinning"
Figure 1
I note that the upper figure is identical to Figure 7a of Danehkar (2022)
l.42 It is arguable whether MUSE is useful for PN kinematics on account of
the low spectral resolution (~2500).
2. Kinematic structures of PNe with [WR] stars
l.59 What does "precision" mean? Is this a confusion for the channel width of
the spectra? The precision with which a line velocity can be measured is
generally much less than the spectral channel width.
l.72 This referee would argue strongly that PV arrays and SHAPE do not put
precise constraints (or even accurate constraints) on the 3D geometry,
since an expansion velocity law must be specified in order to derive the
line-of sight matter distribution from velocity profiles.
l.83 MGC -> NGC
l.83 pointed -> pointed out
l.89 details -> detail
l.105 respected -> respect
l.91 I could not find this density estimate in Danehkar & Parker (2015) and
it indeed seems quite large. What is the reference for the electron density
of Hen 3-1333 and Hen 2-113.
l.113
envelop in a faint halo -> are enveloped by a faint halo
l.123 How does the author know that the inclination of 80deg. applies to
the [N II] and H-alpha emission structures. If the [N II]/H-alpha ratio is
high and the [N II] emission arises in a distinct neutral region, the
inclinations of the H-alpha and [N II] emission regions may differ, and
hence it may not be possible to derive the deprojected velocity for H-alpha.
Figure 3
l.4 of caption
What is "~10 averaged H-alpha SB from SHS"? Is this 10 x 10^-15 ergs cm-2 s-1
per spaxel comparable to the colour bar? If not, some comparison between the two
scales should be included.
l.138 respected -> respect
l.156 What is meant by "the continua in the PV arrays were also resolve .. ".
Continuum cannot be resolved if it is genuinely continuous. Surely the author
means the continua were detected.
l.166
The referee is unclear why higher spectral and spatial resolution can result
in improved nebular inclinations. Usually higher resolutions lead to detection
of substructure which complicates the determination of inclination angles.
Since PNe are not perfect geometric figures, is inclination such an important
defining concept?
l.170
The word morphological is often used for the 2D (on-sky) appearance of a
nebula, whilst I believe the author implies 3D morphology here. I like the
wording "Morpho-kinematic" in the title of Danehkar (2022) as being more
descriptive, but of course it is understandable that the author would not
want to repeat this word in the title of the present paper.
l. 187
The author's claim that the high velocities are not present (slow down?) at
offset > 4", would fit with the suggestion that the supposed high velocities
are only seen where the line is strong and the line wings can be detected
(see Major point 2, above). At larger radial offsets, the nebula brightness
is much lower and no line wings would be expected to be detected.
l.192
The reader is very curious which IFU is planned for resolution much higher
than WiFeS. Currently available IFU's with higher spectral resolution than
WiFeS are FLAMES IFU, which has a small field of view, and MEGARA.
Abbreviations
HWHM should be included in the list.
Author Response
RESPONSES TO THE REVIEWER'S COMMENTS:
COMMENT: This paper presents a summary of kinematic results with WiFeS already published by the author and new data and analysis of H-alpha kinematics of several of these [WR] PNe. The paper needs some minor improvements before being acceptable for publication.
RESPONSE: I have carefully read the reviewer’s comments, and I am thankful to the referee for the helpful suggestions. I have revised the manuscript and addressed the referee's comments as follow:
Major comments
--------------
1. COMMENT: l.62
Basurah et al. (2016, 2016MNRAS.458.2694B) have pointed out the possible problems and ill definition of the wels category of PNe and this should be mentioned.
1. RESPONSE: Added “However, IFU spectroscopy and photoionization modeling of 4 PNe with supposed wels suggested that the wels classification could be spurious, since some assumed stellar emission lines could be of nebular origin and 3 of the presumed wels are hydrogen-rich O(H)-type [52].”
2. COMMENT: l. 145 Sentence beginning "However, the velocity ..." and referring to
Figure 5.
Are the authors sure that these are high velocity line wings, indicative of nebular expansion velocities > HWHM. From the figure, the referee was struck that both PNe have similar width wings, so could these wings be the intrinsic wings of the spectrograph line spread function. Alternatively are these scattering wings of the intrinsic H-alpha emission of the PNe, such as electron scattering, and not necessarily indicative of high velocity outflow. It is not possible to judge how much fainter are the wings than the peaks from Fig. 5, but examination of Fig. 4 suggests a factor > 1000. This point should at least be mentioned.
2. RESPONSE: Revised: “However, the velocity channel maps indicate that the outflow velocities of the faint multipolar lobes could be much higher than the *mass-weighted* HWHM expansion velocities.”
In De Marco et al (1997, MNRAS 292, 86), they mentioned that they “find no detectable amounts of hydrogen in either stellar wind” and and “The lack of evidence for Balmer line emission from the stellar winds”. This point and scattering were clarified in the revised version: “Previously, they found no evidence of Balmer line emission of stellar origin in these objects [46,51], so the Hα emission profiles could be attributed to nebular ionization properties. Nevertheless, the broad H-alpha wings in young, compact PNe may also be created by Rayleigh-Raman scattering [53].”
Minor comments
--------------
1. COMMENT: Abstract
l.12 to read ... lobes which reach ...
1. RESPONSE: Abstract revised.
2. COMMENT: The last sentence is not clear - what further light will higher spectral resolution observations reveal given that the WiFeS observations already have modest spectral resolution.
2. RESPONSE: Revised: “As the WiFeS does not have adequate resolution for compact PNe, future high-resolution spatially-resolved observations are necessary to unveil full details of their morpho-kinematic structures.”
3. COMMENT: 1. Introduction
l.19 Surely the Universe is H-rich. The interest of PNe is that they can be enriched in heavier elements indicative of phases of nuclear reprocessing on the AGB and PN tracks.
3. RESPONSE: Added: “as well as elemental products of nucleosynthesis processing in the AGB stage (e.g. [4–6]).”
4. COMMENT: l.33 suggest "rotating" rather than "spinning"
4. RESPONSE: Corrected.
5. COMMENT: Figure 1
I note that the upper figure is identical to Figure 7a of Danehkar (2022)
5. RESPONSE: Revised a bit.
6. COMMENT: l.42 It is arguable whether MUSE is useful for PN kinematics on account of the low spectral resolution (~2500).
6. RESPONSE: Revised: “enabled the constraint of the ionization structures”.
7. COMMENT: 2. Kinematic structures of PNe with [WR] stars
l.59 What does "precision" mean? Is this a confusion for the channel width of the spectra? The precision with which a line velocity can be measured is generally much less than the spectral channel width.
7. RESPONSE: Corrected: “velocity resolution”
8. COMMENT: l.72 This referee would argue strongly that PV arrays and SHAPE do not put precise constraints (or even accurate constraints) on the 3D geometry, since an expansion velocity law must be specified in order to derive the line-of sight matter distribution from velocity profiles.
8. RESPONSE: Revised: “put constraints on the geometry with specified density and velocity laws”
9. COMMENT:
l.83 MGC -> NGC
l.83 pointed -> pointed out
l.89 details -> detail
l.105 respected -> respect
9. RESPONSE: Corrected.
10. COMMENT:
l.91 I could not find this density estimate in Danehkar & Parker (2015) and it indeed seems quite large. What is the reference for the electron density of Hen 3-1333 and Hen 2-113.
10. RESPONSE: Cited: “( ∼ 10^5 cm − 3 [38]) bodies.”
11. COMMENT: l.113
envelop in a faint halo -> are enveloped by a faint halo
11. RESPONSE: Corrected.
12. COMMENT: l.123 How does the author know that the inclination of 80deg. applies to the [N II] and H-alpha emission structures. If the [N II]/H-alpha ratio is high and the [N II] emission arises in a distinct neutral region, the inclinations of the H-alpha and [N II] emission regions may differ, and hence it may not be possible to derive the deprojected velocity for H-alpha.
12. RESPONSE: Text revised: “at the inclination of 82 ◦ adopted by Danehkar et al. [28]. However, the halos around the knots seem to expand with a higher velocity in the Hα emission.”
13. COMMENT: Figure 3
l.4 of caption
What is "~10 averaged H-alpha SB from SHS"? Is this 10 x 10^-15 ergs cm-2 s-1 per spaxel comparable to the colour bar? If not, some comparison between the two scales should be included.
13. RESPONSE: Corrected: “the boundary of 10 percent of the mean Hα surface brightness of this object in the SHS”
14. COMMENT: l.138 respected -> respect
14. RESPONSE: Corrected.
15. COMMENT: l.156 What is meant by "the continua in the PV arrays were also resolve .. ".
Continuum cannot be resolved if it is genuinely continuous. Surely the author means the continua were detected.
15. RESPONSE: Corrected: “The continua in the PV arrays were also detected and subtracted...”
16. COMMENT: l.166
The referee is unclear why higher spectral and spatial resolution can result in improved nebular inclinations. Usually higher resolutions lead to detection of substructure which complicates the determination of inclination angles. Since PNe are not perfect geometric figures, is inclination such an important defining concept?
16. RESPONSE: Corrected: “Detection of substructures of these compact objects require spatial and velocity resolution higher than the WiFeS.”
17. COMMENT: l.170
The word morphological is often used for the 2D (on-sky) appearance of a nebula, whilst I believe the author implies 3D morphology here. I like the wording "Morpho-kinematic" in the title of Danehkar (2022) as being more descriptive, but of course it is understandable that the author would not want to repeat this word in the title of the present paper.
17. RESPONSE: Changed: “...disclose detailed morpho-kinematic properties...”
18. COMMENT: l. 187
The author's claim that the high velocities are not present (slow down?) at offset > 4", would fit with the suggestion that the supposed high velocities are only seen where the line is strong and the line wings can be detected (see Major point 2, above). At larger radial offsets, the nebula brightness is much lower and no line wings would be expected to be detected.
18. RESPONSE: Revised. “while the nebular emission is weakened as approaching the radial distance of 8′′. ... However, we caution that the broad Hα wings in these compact PNe could be associated with Rayleigh-Raman scattering [55].”
19. COMMENT: l.192
The reader is very curious which IFU is planned for resolution much higher than WiFeS. Currently available IFU's with higher spectral resolution than WiFeS are FLAMES IFU, which has a small field of view, and MEGARA.
19. RESPONSE: Revised as it could be long-slit: “Detailed morpho-kinematic modeling of these compact PNe can be performed when kinematic observations with spatial and wavelength resolution higher than the WiFeS are available.”
20. COMMENT: Abbreviations
HWHM should be included in the list.
20. RESPONSE: Added: “the half width at half maximum (HWHM)”
Reviewer 4 Report
The work presents results of IFU data taken in 2010. A few papers have been previously published by the author based on this data and this work focuses on the Halpha emission line results from that data. A good fraction of the work mentions results from the previous papers trying to put the Halpha results into context. It is peculiar that this analysis was not performed in previous works. From the results presented here it seems that no new relevant information is gained from the Halpha maps. It seems to me that the focus should have been more on the utility of small telescope IFUs for the field rather than on pushing the limits of detection in Halpha maps. Given that it is very difficult to get large telescope time to observe PNe with IFUs, the instrument used here can fill an important niche. It would have been nice to see this point made more clear. In general I think the paper is ok for a conference contribution. I have made a few more detailed comments below.
- What are the errors on estimated velocities? They should appear in the abstract and in the text in general.
- There is no description of the reduction process and steps for the data. This should appear somewhere.
Figure 2 does not really give any relevant information. The slices do not show any information about the velocity structure. I recommend this be removed or replaced by 2d position velocity slices that can show more clearly the information.
- it is hard to judge how good the models in Figure 1 are without having the observational velocity maps to compare to. The author could add a third line in the lower panel of Figure one with the observed velocity map. This would help this work be more self contained. Looking at the images in the reference cited, it seems to me that for some objects the velocity maps do not have enough velocity resolution to constrain the model. Perhaps the author should comment in the text more clearly the limitation of the data.
L84-85 - I think this statement is a bit strong given the quality of the data in the reference. However since the data is not shown here I suggest the author remove the phrase.
l95 - This phrase should be revised. I think perhaps the reference comes before the D=9.44 part?
L103 - why not show the [NII]6584 channel maps? Or perhaps both? The [NII] maps should in principle show more detail since it is typically formed in the outer parts of the nebula.
L113 - are enveloped by the halo
L120 - If the author is going to mention all the results from the previous work, it would be nice to see some of that in Figure 3. Perhaps overplot [NII]contours so we can see how the low ionization region compares.
Wouldn't a PV map of the Halpha line be more useful than the channel maps in figure 3 since the objective is to show velocities?
L143 - define HWHM
I recommend that the author make a thorough English revision of the manuscript.
Author Response
RESPONSES TO THE REVIEWER'S COMMENTS:
COMMENT: The work presents results of IFU data taken in 2010. A few papers have been previously published by the author based on this data and this work focuses on the Halpha emission line results from that data. A good fraction of the work mentions results from the previous papers trying to put the Halpha results into context. It is peculiar that this analysis was not performed in previous works. From the results presented here it seems that no new relevant information is gained from the Halpha maps. It seems to me that the focus should have been more on the utility of small telescope IFUs for the field rather than on pushing the limits of detection in Halpha maps. Given that it is very difficult to get large telescope time to observe PNe with IFUs, the instrument used here can fill an important niche. It would have been nice to see this point made more clear. In general I think the paper is ok for a conference contribution. I have made a few more detailed comments below.
RESPONSE: I have carefully read the reviewer’s comments, and I am thankful to the referee for the helpful suggestions. I have revised the manuscript and addressed the referee's comments as follow:
1. COMMENT: - What are the errors on estimated velocities? They should appear in the abstract and in the text in general.
1. RESPONSE: Included: “\pm 20”
2. COMMENT:- There is no description of the reduction process and steps for the data. This should appear somewhere.
Figure 2 does not really give any relevant information. The slices do not show any information about the velocity structure. I recommend this be removed or replaced by 2d position velocity slices that can show more clearly the information.
2. RESPONSE: The reduction process and steps are given in detail in Ref. [29]. This figure illustrate PPV cubes, and 2d position velocity slices are presented in Danehkar [29] and Fig. 1.
3. COMMENT:- it is hard to judge how good the models in Figure 1 are without having the observational velocity maps to compare to. The author could add a third line in the lower panel of Figure one with the observed velocity map. This would help this work be more self contained. Looking at the images in the reference cited, it seems to me that for some objects the velocity maps do not have enough velocity resolution to constrain the model. Perhaps the author should comment in the text more clearly the limitation of the data.
3. RESPONSE: Mentioned: “Redshift and blueshift effects with respect to the nebula center are rendered by red and blue colors, which can be compared with the projected velocity IFU maps (Fig. 1 in Danehkar [29]).” The observed velocity maps will be published by Ref. [29] in ApJS, and are protected by the AAS copyright. Velocity resolution and seeing were mentioned in Ref. [29].
4. COMMENT:L84-85 - I think this statement is a bit strong given the quality of the data in the reference. However since the data is not shown here I suggest the author remove the phrase.
4. RESPONSE: Text revised.
5. COMMENT:l95 - This phrase should be revised. I think perhaps the reference comes before the D=9.44 part?
5. RESPONSE: The references of other distances were cited in Danehkar et al. (2016, AJ, 151, 38). We cited the most recent one in this paper.
6. COMMENT:L103 - why not show the [NII]6584 channel maps? Or perhaps both? The [NII] maps should in principle show more detail since it is typically formed in the outer parts of the nebula.
6. RESPONSE: The [N II] channel maps were shown in Danehkar et al. (2016, AJ, 151, 38), and are protected by the AAS copyright.
7. COMMENT:L113 - are enveloped by the halo
7. RESPONSE: Corrected.
8. COMMENT: L120 - If the author is going to mention all the results from the previous work, it would be nice to see some of that in Figure 3. Perhaps overplot [NII]contours so we can see how the low ionization region compares.
8. RESPONSE: The [N II] channel maps were published in Danehkar et al. (2016, AJ, 151, 38).
9. COMMENT:Wouldn't a PV map of the Halpha line be more useful than the channel maps in figure 3 since the objective is to show velocities?
9. RESPONSE: We decide to present the H-alpha channel maps for comparison with the [N II] channel maps presented in https://arxiv.org/abs/1601.01702.
10. COMMENT:L143 - define HWHM
10. RESPONSE: Added: “the half width at half maximum (HWHM)”
Round 2
Reviewer 1 Report
Although this paper has been heavily modified in the revised version, few minor issues are needed to be clarified in the current draft. This paper can be considered to publish after minor modifications. Few comments and suggestions are listed below:
Comments:
The modifications presented in current draft are suggested to mark as "boldface" for displaying clearly.
Some symbols listed in the draft are found to be inconsistent such as "Figure" and "Fig.".
1. Section 2, Paragraph 1:
".... Galactic bulge according to D = 9.44..."
What is the unit of the value of 9.44?
2. Discussion:
1. Paragraph 3 and 4:
"Figure 4 presents ... cubes of Hb 4 and M 2-42 as shown in Fig. 5"
Have these 3D models (Figure 4) and PPV cubes (Figure 5) been constructed after considering extinction effect? If not, why?
Author Response
COMMENT: Although this paper has been heavily modified in the revised version, few minor issues are needed to be clarified in the current draft. This paper can be considered to publish after minor modifications. Few comments and suggestions are listed below:
Comments:
The modifications presented in current draft are suggested to mark as "boldface" for displaying clearly.
RESPONSE: Changes highlighted.
1. COMMENT: Some symbols listed in the draft are found to be inconsistent such as "Figure" and "Fig.".
1. RESPONSE: Corrected. “Fig.” changes to “Figure”.
2. COMMENT: 1. Section 2, Paragraph 1:
".... Galactic bulge according to D = 9.44..."
What is the unit of the value of 9.44?
2. RESPONSE: Added: “D = 9.44 kpc”
3. COMMENT: 2. Discussion:
1. Paragraph 3 and 4:
"Figure 4 presents ... cubes of Hb 4 and M 2-42 as shown in Fig. 5"
Have these 3D models (Figure 4) and PPV cubes (Figure 5) been constructed after considering extinction effect? If not, why?
3. RESPONSE: Correcting extinction is not required when only one emission line is involved in the study and our aim is morpho-kinematic analysis, and we are not interested in physical conditions and chemical abundances.
Reviewer 2 Report
I have one remaining point. Line 102 to 107 seems to claim that the NII knots decelerate with higher velocity gas in front of them. That cannot be right? If they are decelerating, the gas in front that causes the friction should be slower, not faster.
Author Response
COMMENT: I have one remaining point. Line 102 to 107 seems to claim that the NII knots decelerate with higher velocity gas in front of them. That cannot be right? If they are decelerating, the gas in front that causes the friction should be slower, not faster.
RESPONSE: Revised: “The higher outflow velocities seen in the Hα channels could mostly be associated with the tenuous halos *around* the two detached N + low-ionization point-symmetric knots escaping from the core with slower outflow velocities.”
Reviewer 4 Report
Although I think this manuscript could be significantly improved, it is sufficient for a proceedings book, given the limitations.
Author Response
COMMENT: Although I think this manuscript could be significantly improved, it is sufficient for a proceedings book, given the limitations.
RESPONSE: No comments.